# Comparison of Imaging Methods and Population Pattern in Dogs with Spinal Diseases in Three Periods between 2005 and 2022: A Retrospective Study

**DOI:** 10.3390/vetsci10050359

**Published:** 2023-05-18

**Authors:** Jakub Fuchs, Michal Domaniža, Mária Kuricová, Tomáš Lipták, Valent Ledecký

**Affiliations:** Small Animal Clinic, Veterinary University Hospital, University of Veterinary Medicine and Pharmacy, Komenskeho 73, 04181 Kosice, Slovakia; jakub.fuchs@student.uvlf.sk (J.F.); michal.domaniza@student.uvlf.sk (M.D.); tomas.liptak@uvlf.sk (T.L.); valent.ledecky@uvlf.sk (V.L.)

**Keywords:** canine, epidemiology, imaging methods, retrospective study, spinal cord

## Abstract

**Simple Summary:**

Imaging methods are an essential component in the diagnosis of spinal disorders. Plain and contrast radiographs are widely used in small animal clinical practice; however, the increasing availability of sophisticated imaging methods such as computed tomography and magnetic resonance imaging (MRI) is improving the approach to patient care. Our hypothesis was that the availability of MRI has increased over the years, increasing the diagnostic and therapeutic success rates. We compared the use of two imaging methods (X-ray myelography, MRI) over a period of time and the occurrence of neurologic disease diagnosed with each modality. We found that the therapeutic success rate increased as the number of dogs undergoing MRI increased.

**Abstract:**

The aim of this study was the long-term comparison of the imaging methods used in dogs with neurologic diseases related to the spine and spinal cord. We also compared the occurrence of neurological diseases according to the localization, gender, age, and breed. As the availability of magnetic resonance imaging (MRI) has increased over the years, resulting in increased diagnostic and therapeutic success rates, the study was divided into three time periods (2005–2014, 2015–2018, and 2019–2022). Our results suggest changes in the population structure of the dogs studied and changes in the use of diagnostic methods that directly or indirectly influence the choice and success rate of therapy. Our results may be of interest to owners, breeders, practicing veterinarians, and insurance companies.

## 1. Introduction

Neurological diseases originating from the affected spinal cord are one of the most challenging clinical cases in veterinary practice and may occur due to different reasons such as abnormalities of anatomy, infections and inflammations, degenerative disorders, or traumatic causes [1,2]. The most common causes of spinal cord injury in dogs include degenerative diseases, including intervertebral disc herniation, spondylosis deformans, and degenerative lumbosacral stenosis, followed by traumatic injuries (vertebral fractures or luxations, concussions, compressions or lacerations of the spinal cord), neoplasia, and fibrocartilaginous embolism. The most common immune-mediated condition is steroid-responsive meningitis–arteritis, and the most common infectious condition is discospondylitis [2,3]. Moreover, the influence of weather conditions, environmental, and lifestyle factors such as the influence of diet and the level of physical activity on the development of various diseases has also been investigated in humans [4,5,6]. In one study on Dachshunds, the authors demonstrated that dogs with an increased level of exercise, more than 1 h per day, were less likely to develop intervertebral disc disease (IVDD) [7]. In another study regarding dogs by Barandun et al. (2020), the authors described the effect of cold temperatures on the mobility of the musculoskeletal system, whereas it also notably affected the spinal column. With both reduced mobility and resistance to strain injuries, biomechanical forces could act differently or more traumatically on the spine, although this is purely speculative [8].

In addition to signalment and anamnesis, the diagnostics of neurologic diseases affecting the spine and spinal cord is mostly based on the findings of neurologic examinations and imaging methods. Radiographic evaluation is based on lateral and ventrodorsal orthogonal projections, these are performed after stabilization of the animal in cases of trauma. During the entire examination, it is very important to minimize the manipulation with the spinal column [9]. Therefore modern veterinary neurology and neurosurgery cannot exist without diagnostically very reliable and sophisticated techniques that include computed tomography (CT) and magnetic resonance [10]. Magnetic resonance imaging (MRI) is increasingly used in the diagnostics of small animal spinal cord trauma to identify the site of compression and parenchymal changes [11,12,13,14]. Compared to a CT scan or X-ray imaging, MRI provides more details about the severity, exact localisation, and nature of spinal cord compression. The incidence of canine acute myelopathies has been well documented using MRI [12,15,16,17,18,19]. Progress and an increased availability of the new sophisticated imaging diagnostic methods have created better conditions for veterinary practitioners in terms of diagnostics and therapy of many neurological cases [19].

For the best possible assessment and comparison of diagnostic methods and therapies for serious spine and spinal cord injuries, long-term studies focused on the objective evaluation of diagnostic methods and therapy success rates on larger canine populations are needed. Such studies using extensive databases of clinical records are rarely accessible; however, they are considered very valuable due to the amount of data provided and the expected clinical relevance of the results [20]. Previous studies aimed at a long-term follow up of diseases in canine populations were mostly focused on the evaluation of only one breed of dog (breed disposition) or just one disease (descriptive case series) [21,22,23,24,25]. Few long-term studies have analysed the time trends and risk factors for various diseases in dogs, such as diabetes mellitus from 1970 to 1999 [26], acquired myasthenia gravis from 1991 to 1995 [27], and causes of death or reasons for euthanasia in military working dogs between the years 1993 and 1996 [28]. In a large retrospective study of diagnoses in relation to the gender, age, and breed of dogs in Sweden in 1996, it was found that the skin represented the most frequently affected system, followed by the gastrointestinal tract and the genital system [20]. The authors of one retrospective study created and defined a new database based on canine neurological diseases, with specific variables and a diagnosis key list ‘VITAMIN D’. This study included 4497 dogs of 187 different breeds with a well-documented neurological disease. IVDD was the most frequent diagnosis of all cases (21%), followed by idiopathic epilepsy (8.2%), degenerative lumbosacral stenosis (6.3%), lesions due to fracture/luxation (5%), and space-occupying lesions (3.7%) [29]. Another retrospective study between the years 2002 and 2016 focused on the prevalence of neurologic disorders in 533 French Bulldogs. In 45.5% of all dogs, the most common neurologic disorder was Hansen’s type I intervertebral disc herniation (IVDH), which, in most cases, was located in the thoracolumbar segment (60.2%) and the cervical segment (39.8%) [30]. The thoracolumbar segment represents the most affected segment in most studies regarding dogs [29,31]. In a study comprising 54 dogs with acute thoracolumbar disc herniation and 16 clinically normal dogs, the authors examined the association between cerebrospinal fluid (CSF) biomarkers, initial neurologic dysfunction, and ambulatory outcome of treated dogs [32]. A study by Havig et al. (2005) aimed to evaluate the long-term neurological outcomes in 19 dogs with atlantoaxial subluxation treated non-surgically with a cervical splint. In 10 out of 16 dogs (62.5%), a good final outcome was reported [33]. In one study comparing the conservative and surgical therapy strategies for cervical spondylomyelopathy in 157 dogs, the authors found that a favourable outcome was associated with non-surgical therapy in 54% of dogs and with surgical therapy in 81% of dogs. However, the difference was not statistically significant, probably because of the low patient number [34].

There is only one study focusing on the correlation between MRI and radiographic findings with clinical prognosis in canine spinal cord diseases. In this study, 60 dogs with neurologic symptoms were evaluated, and the authors described that there were no significant association between age, sex, and breed, and the frequency of the IVDD. The sensitivity and specificity of radiography were evaluated as 90.0% and 46.0%, respectively, whereas the MRI was considered as a gold standard modality. There was a significant association between the severity of disease visible on MRI with referral to surgery and medical therapy. The recovery rate after surgery was significantly higher than medical therapy in their study [1].

The primary objective of our study was to compare the use of imaging methods and the success rate of both medical and surgical therapy in different time periods. The secondary objective of our study was to evaluate the possible epidemiological changes of the affected canine population with spinal cord dysfunction according to gender, body weight, age, breed, localization of the lesion, and seasonal occurrence for three time periods between the years 2005 and 2022.

## 2. Materials and Methods

### 2.1. Imaging Protocols

Myelographic examination was performed after radiography under general inhalation anaesthesia following the standard protocol (usually cranial myelography followed by caudal puncture). Iohexol (Omnipaque 350 mgI/mL GE Healthcare, Oslo, Norway) was used in all dogs at a dose of 0.3 to 0.5 mL/kg. After contrast injection, standard orthogonal radiographs of cervical, thoracic, and lumbar segments were obtained immediately, and 5, 10, 15, and 30 min after injection as needed for individual patients and depending on the flow of the contrast agent. 

All MRIs were performed using a GE Signa Explorer 1.5 T scanner with USCT234 posterior array coil to acquire standard sequences with slice thickness of 3 mm and GAP of 0.3 mm. The standard protocol included T1W sagittal, T2W sagittal, T2W dorsal, T1W transverse, T2W transverse, STIR sagittal, T1W sagittal post-contrast, T1W transverse post-contrast, and additional sequences such as T2* (gradient echo) transverse or T1W post-contrast FAT SAT if needed. The gadolinium (Clariscan 0.5 mmol/mL, GE Healthcare Oslo, Norway) dose was 0.2 mL/kg of body weight.

### 2.2. Data Collection

We performed a retrospective analysis of spinal diseases occurrence and imaging methods use in dogs between the years 2005 and 2022. All cases included in this study were referral patients from veterinary practitioners, and all data were obtained from the electronic patient database. Cases with definitive or presumptive diagnosis related to the spine in which the clinical diagnosis was supported by a complete neurologic examination by an experienced university clinician and other diagnostic measures (radiographs, MRI, and CSF analysis) were included. Dogs for which gender, age, body weight, breed, imaging method used, method of therapy, and outcome were recorded were included in this study. Patients with incomplete records were excluded from the study (9% of dogs). After data collection, we evaluated the number of dogs enrolled in each month throughout the period between 2005 and 2022 to determine any seasonal predispositions. 

We divided the study into three time periods. The first period was characterized by low use of MRI, whereas the main diagnostic procedures consisted of radiographs and myelography (2005–2014). In the second period, both radiographs and myelographs were performed, with myelography predominating; however, few patients also underwent MRI (2015–2018). In the third period, most dogs underwent MRI, although radiographs continued to be performed with myelography (2019–2022). These periods were chosen because the number of dogs examined with MRI gradually increased and the dog population changed in recent years in terms of breed distribution.

The clinical database of our hospital was searched for patient registration and appropriate keywords using ProVet clinical software. When information was missing from the electronic records, paper records were retrieved when available. Information on gender, age, body weight, breed, duration of clinical signs, neurologic dysfunction score (MFS—modified Frankel score) [35], imaging method, lesion localization, diagnosis, therapy method, and outcome was collected, and complete data were exported to Excel 2020 (Microsoft Office 2020), where charts were also created. Univariate analysis was used for statistical analysis. The obtained data are expressed as Median, Min–Max, and Mean ± SD. Differences between groups assessed in 3 time periods were analysed using multiple *t*-tests. Differences were considered statistically significant when *p* < 0.01. Chi-square test of independence was used to compare the frequency of the use of different imaging techniques in the evaluated time periods. Differences were considered statistically significant when *p* < 0.05.

## 3. Results

During the studied period, a total of 29,800 patients visited our workplace, of which 998 dogs were neurological patients with affected spines. This study included 597 dogs between the years 2005 and 2014, 204 dogs between 2015 and 2018, and 197 dogs between 2019 and 2022.

### 3.1. Gender, Age, Body Weight, Breed

During the initial study period from 2005 to 2014, 597 dogs were presented with a presumptive spinal diagnosis, an average of 54 dogs per year. The proportion of male dogs was 58%, and the proportion of female dogs was 42%. 

During the second period from 2015 to 2018, 204 dogs were enrolled in the study, representing an average of 51 dogs per year. Of these, 42% were males and 58% were females. 

The third period between 2019 and 2022 included 197 dogs, an average of 56 dogs per year. The proportion of male dogs was 48%, and the proportion of female dogs was 52%. The gender, age, and body weight of the population are shown in Table 1.

Table 1 shows that the dogs in the first studied period had significantly higher bodyweights than the groups of dogs in the following time periods. The changing trend in breed distribution is shown in Table 2 as the number and percentage of breeds most affected during in each time period studied.

Between 2005 and 2014, the other breeds (*n* = 296) consisted of 12 different breeds, ranging from miniature Maltese dog (1.2 kg) to giant Great Dane (65 kg), with small and miniature breeds accounting for 15% of this group. 

Between 2015 and 2018, the other breeds group (*n* = 93) consisted of nine different breeds, ranging from the smallest Yorkshire Terrier (1.3 kg) to the largest Great Dane (54 kg), while small and miniature breeds accounted for 20% of the group.

Between 2019 and 2022, the other breeds group (*n* = 94 dogs) was represented by 16 different breeds, including the smallest dog of an unspecified breed (1.6 kg) and the largest being a Greater Swiss Mountain Dog (59 kg), while small and miniature breeds accounted for 29% of this group. 

### 3.2. Diagnostic Imaging

During the first study period (2005–2014), most of the dogs only underwent radiographic imaging (*n* = 382), including 113 dogs with visible pathologic changes on plain radiographs. Myelography was performed in 169 dogs, and a diagnosis was made in 79 dogs (47%), which were subsequently treated surgically. MRI was performed in only three cases (0.5%). The owners of the remaining dogs (*n* = 212, 35%) with obvious clinical signs of neurologic dysfunction originating from the affected spinal cord had refused further investigations.

The majority of dogs during the second study period (2015–2018) were diagnosed by myelography (*n* = 103), followed by MRI (*n* = 30). The remaining dogs (*n* = 71) were diagnosed by neurologic examination and/or radiographs only, and their owners declined further examination.

During the third study period (2019–2022), the majority of dogs were diagnosed with MRI (*n* = 93), whereas 20 of them had MRI performed after myelography failed to reveal a diagnosis. Myelography was performed in 54 dogs, and a diagnosis was made in 34 (63%) cases. In the remaining 50 dogs (25.5%), no further diagnosis was made because their owner declined examination. 

Throughout the whole study period, the occurrence of negative side effects of myelography was reported, such as brief seizures, compulsive movements, and spasms (3.3% of dogs, *n* = 10/306).

The percentage of different imaging methods during the study period is shown in Figure 1. The frequency of radiographic examination was significantly higher in the first study period (*p* < 0.05) than in the second and third periods. Radiography was the most frequently used diagnostic tool throughout the study. The frequency of MRI examination was significantly higher (*p* < 0.05) in the third evaluated period than in the previous two periods.

### 3.3. Neuroanatomic Localization

During the first study period (2005–2014), most dogs underwent radiographic examination only after completion of neurologic, clinical, and haematological/biochemical examinations. Most diagnoses were assigned to the thoracolumbar spinal segment T3-L3, followed by the lumbosacral segment L4-S3, the cranial cervical segment C1-C5, and the cervicothoracic segment C6-T2. Neuroanatomic localization could not be confirmed in all dogs, and most diagnoses were presumptive (50%). They were classified as degenerative in 35% of cases, inflammatory or infectious in 20%, traumatic in 12%, neoplastic in 8%, and other in 16% of cases (vascular, metabolic/toxic, anomalous). 

In 204 dogs examined between 2015 and 2018, the most commonly affected spinal cord segment was the T3-L3 segment, followed by the L4-S3 segment, the C1-C5 segment, and the C6-T2 segment. The diagnoses were degenerative in 46% of cases, inflammatory/infectious in 19%, traumatic in 13%, neoplastic in 8%, metabolic/toxic in 6%, vascular in 6%, and anomalous in 3% of cases. 

During years 2019 and 2022, the most commonly affected spinal cord segment in the 197 dogs examined was also the T3-L3 segment, followed by the cervical segment C1-C5, the lumbosacral segment L4-S3, and the C6-T2 segment. The diagnoses were degenerative in 38%, traumatic in 19%, inflammatory/infectious in 10%, vascular in 10%, neoplastic in 9%, metabolic/toxic in 8%, and anomalous in 7% of cases.

A summary of the affected spinal cord segments with the most affected breed in each case can be found in Table 3. The specific diagnoses identified throughout the study period according to the differential list VITAMIN D are summarized in Table 4.

Some of the reported diagnoses are shown in Figure 2, Figure 3, Figure 4, Figure 5, Figure 6, Figure 7 and Figure 8.

### 3.4. Therapy

Because of the lack of definitive diagnoses in the first study period between 2005 and 2014, and because most diagnoses were based on radiographs and clinical/neurologic examinations, therapy was mostly symptomatic and conservative (73%), and some owners opted for euthanasia because of poor prognosis or financial cost (14%). Surgical therapy was performed for severe or progressive neurologic deficits, i.e., limb paralysis, flaccid paresis, and urinary incontinence with proven compressive spinal cord lesions in 13% of dogs. Hemilaminectomy was performed in 45 dogs (with surgical stabilization in 5 dogs), laminectomy in 25 dogs, and ventral slot in 9 dogs (with surgical stabilization in 2 dogs). During this period, surgical therapy provided better results than conservative therapy, with success rates of 80% and 54%, respectively.

From 2015 to 2018, a total of 24% of dogs (*n* = 49 of 204) were surgically treated with hemilaminectomy (*n* = 30), dorsal laminectomy (*n* = 2), and ventral slot (*n* = 17). Vertebral stabilization was performed in two dogs. The dogs were representatives of 13 breeds and mixed breeds, with the male gender being over-represented (28:21). During this period, conservative therapy was performed in 135 dogs (66%), and the remaining dogs were humanly euthanised because of poor prognosis due to lack of nociception (*n* = 20, 10%). During this period, surgical therapy had significantly better results than medical therapy, based on the faster and better improvement of MFS postoperatively (mean MFS improved by 1.6 degree 2 days after surgery, versus 0.3 in conservatively treated dogs, *p* < 0.05). In addition, conservative therapy was associated with more frequent recurrences (62% vs. 30%). MFS was evaluated 3 months after therapy, and maximum MFS was higher in surgically treated dogs (1.86 and 1.0 respectively, *p* < 0.05).

Surgical therapy was performed in 111 dogs (56.4%) between 2019 and 2022 and included hemilaminectomy (*n* = 78) with stabilization in 10 dogs, dorsal laminectomy (*n* = 13), and ventral slot (*n* = 20) with stabilization (*n* = 6 of 20). Humane euthanasia was performed in 13 dogs (6.5%) at the owners’ request. Conservative therapy was used in 73 dogs (37.1%). During this period, surgical therapy achieved significantly better results; it shortened the time required for improvement in neurologic deficits and resulted in earlier recovery. Surgically treated dogs had higher MFS approximately 14 days earlier. The first improvement was recorded from the first day after surgery, compared with conservative therapy, in which the first improvement occurred on average after 5 days, and was associated with a higher rate of recurrence. 

### 3.5. Seasonal Occurrence

In the present study, we analysed the number of affected dogs in each month of the year. The months with the highest incidence were February, March, April, May, October, November, and December. The number of dogs in each month varied greatly, as can be seen in Figure 9.

## 4. Discussion

We found that during years 2019–2022, the frequency of MRI use increased significantly by more than 30% compared to 2015–2018. The use of radiographs as the sole diagnostic imaging tool decreased significantly by 39% in 2019–2022 compared to 2005–2014. The use of conservative treatment did not decrease significantly over time, while we found that the success rate of therapy increased by 20% in 2015–2018 and showed the same trend in subsequent years. We reported a more frequent occurrence of spinal disorders in small dog breeds, while the chondrodystrophic breeds were most frequently affected. The most common diagnoses were classified as degenerative and traumatic, and the most commonly affected spinal cord segment was the T3-L3 segment. No gender predisposition was found. However, the exact incidence of spinal cord disease in dogs is not known. Nevertheless, it is estimated that 2% of all patients admitted for therapy suffer from IVDD and 60% of patients with spinal cord injury are car accident cases [36,37]. We reported the incidence of spinal cord diseases in 3% of all dogs admitted to our clinic. The predominant causes of spinal cord injury are degenerative diseases, especially IVDD, which affect the spinal cord and account for the majority of diagnoses in the canine populations studied. Car accidents as a known cause of trauma were reported in 22% of all traumatic diagnoses. In a study by Barandun et al. (2020), the results were consistent with those of human studies showing that lower ambient temperatures were associated with more pain and increased risk of muscle injuries, which is similar to our findings. Most dogs were admitted in the months of February, March, April, and May, followed by the autumn and winter months, when the ambient temperature tends to be lower. This is leading to spasms and changes in biomechanical forces [8], but the lower occurence may also be due to the fact that most people spend vacations during summer and warm months, and therefore, the number of recorded cases in dogs is lower. The effect of temperature on the occurrence of intervertebral disc disease observed in this study is also found in some human studies on a number of diseases, where lower ambient temperatures are associated with more pain in conditions such as osteoarthritis, pelvic pain syndrome, and musculoskeletal pain, and may also increase the risk of muscle injury. However, in contrast to the findings in dogs, a similar study in humans found no association between the frequency of hospitalisation for back pain and different weather conditions [38]. In the diagnosis of neurologic diseases originating from the affected spine, radiography is still the most commonly used method because of its wide availability. In the case of spinal cord compression, myelography has a higher predictive value for lesions in the thoracolumbar segment than in the lumbosacral segment of the spinal cord. However, advanced imaging techniques are often required to confirm and specify the complete distribution of extruded disc material within the spinal canal. Plain radiography performed under general anaesthesia prior to myelographic examination can reveal the localisation of primary compression in 51–61% of cases [39,40]. MRI has been shown to be more predictive than CT or myelography, but no explicit studies have been performed comparing the progression of the most commonly used imaging techniques over the years. It has been shown that there is no correlation between the degree of compression detected by MRI or myelography and the clinical manifestation of neurologic deficits [41]. Similarly, the degree of compression detected by MRI does not correlate with the speed of onset and duration of clinical signs or with the outcome of surgical therapy, with no difference between chondrodystrophic and nonchondrodystrophic breeds. Therefore, the degree of compression as a prognostic indicator is highly controversial [42,43]. In addition, myelography is an invasive technique with the risk of various side effects such as seizures, worsening of neurological symptoms, asystole, and renal failure [44], but it is still widely used and continues to be used, which is why it is acceptable for compressive spinal cord lesions and is the only diagnostic method in many locations. The risk of complications after myelography is estimated at 10–20% [19]. In our study, we reported the occurrence of adverse effects after myelography in 10 dogs (3.3%) in the form of short-lasting seizures, compulsive movements, and spasms.

We reported that the number of dogs undergoing different imaging procedures has changed over the years, which is related to the increasing availability of MRI and the increasing willingness of owners to undergo this procedure, recognising its incomparably higher diagnostic and prognostic value. Between 2005 and 2018, myelography was performed three to four times more frequently than MRI. Later, between 2019 and 2022, MRI was used twice as often as myelography. This led to the detection of a greater number of spinal cord lesions, especially parenchymatous, and also to better decision making regarding surgical therapy by both owners and veterinarians in practice. The secondary effect of MRI availability is a higher success rate of therapy. In confirmation of parenchymatous lesions and intervertebral discs herniations, MRI is incomparably better than myelography or CT [45]. Despite its high potential, it remains less accessible in clinical practice of companion animals.

In a study by Fleuhman et al. (2006), the authors found that the most frequently observed breeds were mixed breed dogs (13%), followed by German Shepherd Dogs (10%), Dachshunds (9%), and Pekingese (6%) [29]. In our study, the most frequently observed breed in the first study period was the German Shepherd Dog (20.9%), in the second period the Dachshund (17.1%), and in the third period the French Bulldog (17.2%). Chondrodystrophic breeds and German Shepherd Dogs are among the most frequently treated dogs in our country. Dachshunds and French Bulldogs are mainly found in the first group, and a large number of studies have been dedicated to them worldwide [8,30]. In the study by Filho and Selmi (1999), German Shepherd Dogs were the most frequently affected, followed by Dachshunds, Cocker Spaniels, and cross breeds [46]. Dachshunds were over-represented (35%) in the previously cited studies, suggesting that they may be the most predisposed breed among chondrodystrophic breeds. In addition to anatomical and breed predispositions, there are some controversial theories about the development of IVDD, in which age, sex, body weight, body condition score, muscle mass, and activity play a role in influencing the disease [47,48,49,50]. Our study was composed of 14.3% Dachshunds, 8% French Bulldogs, and 6% Cocker Spaniels. In most studies dealing with neurologic dysfunction, cross breeds were the most commonly affected, followed by German Shepherd Dogs [51]. In addition, in a study by Moore et al. (2001), 15.6% of military working dogs were euthanized due to spinal cord or cauda equina disease [28]. This trend in breed distribution was also found in our study. However, in recent years, there has been a marked increase in the number of miniature and small breeds of dogs evaluated. This may be related to the general increase in these breeds, as they are more popular and are therefore bred more frequently, or to the fact that most of these dogs have no proof of origin (pedigree) and are in fact cross breeds that may have genetically accumulated defects caused by possible inbreeding.

Few studies reported an increased incidence of diseases such as intervertebral disc disease in middle-aged dogs [52,53], which was confirmed by the age distribution of the individual cases in the present study, as the median age of the affected dogs was 83 months (IQR). The median ages for the cervical and thoracolumbar segments of IVDH were 4.2 and 4 years, respectively [30]. The highest incidence for the onset of clinical signs (45% were six to nine years of age) was consistent with findings in the literature [48,54,55]. Compared to previous studies, the age of surgically treated dogs was lower in our case, which may be related to the large number of chondrodystrophic dog breeds in our study, in which the described age at onset of neurologic deficits originating from the spine and spinal cord is lower. Neurologic deficits were observed more frequently in chondrodystrophic breeds than in large, nonchondrodystrophic breeds [51,56,57], which is consistent with our results. The prevalence of clinically affected brachycephalic dogs with so-called screwed tails with congenital vertebral malformations is not precisely known [58,59,60,61]. We reported vertebral anomalies in up to 3% of neurologically affected dogs.

In one study, a total number of 946 cases of IVDD were found. Five hundred and ninety-six dogs with thoracolumbar localization were significantly over-represented [62]. L3-S3 localization accounted for 9% of cases, which is higher than previous results [56,62]. Thoracolumbar lesions accounted for 84–86% of cases, and small dog breeds were most commonly affected, especially Dachshund, Shih Tzu, Lhasa Apso, Welsh Corgi, Pekingese, Maltese, Yorkshire Terrier, and Beagle [63]. The predisposition to disc herniation likely reflects the biomechanical forces related to body structure and genetic factors that influence disc degeneration. Basset Hounds, German Shepherds Dogs, Labrador Retrievers, and Dobermans are the most commonly affected large dog breeds. In a study of 8117 cases of the disease, Dachshunds were found to be 9.9 times more likely to develop intervertebral disc disease than all other breeds [25,40,55,64,65,66,67]. The results of one study showed, that the most common localization of IVDD was between second and third lumbar vertebrae, and between 13th thoracic and first lumbar vertebrae. The most commonly affected breed was a mixed-breed Terrier [1].

Lesions involving the cervical segment discs account for 14% to 25% of all IVDD in dogs. According to several authors, most dogs with IVDD suffer from extrusion rather than disc protrusion. Dogs between 4 and 8 years of age are most commonly affected, and according to others, between 2 and 7 years [68]. Chondrodystrophic and other small breeds are most at risk, with Dachshunds, Miniature Poodles, and Beagles accounting for up to 80% of cases. Medium-sized Poodles, Spaniels, Shih Tzu, Pekingese dogs, and Chihuahuas are also commonly affected. Large dog breeds such as Labrador Retrievers, Dalmatians, and Dobermans can account for up to 24% of all cases [37,62,69,70]. Cervical (C1-T2) localization was diagnosed in 21% of cases, which is slightly lower than previous results [48]. An almost even distribution was found between C1-C5 (10%) and C6-T2 (11%) localization [56,62]. As can be seen from our results, we found some differences between the time periods based on the neuroanatomical localization of the lesions. In the studied periods, the most frequently affected segment was the thoracolumbar segment (56.3%). The second most frequently affected spinal cord segment between 2015 and 2018 was the lumbosacral segment (40.7%) and the cervical segment (24.8%). Over the entire evaluated period (2005–2022), the cervicothoracic (C6-Th2) segment was the least affected (<6%).

Treatment of dogs with spinal cord injury usually consists of stabilizing of post-traumatic conditions by physical and pharmacologic methods, and by surgical decompression of the spinal cord. Successful therapy of dogs with neurologic deficits is based on surgical decompression of the affected segment, followed by surgical stabilization when needed [71]. Despite some positive results described after conservative/medical therapy, surgical therapy is the most effective choice of therapy [72,73]. In mild cases of IVDD, when the owner cannot afford surgery or surgery would pose a life-threatening risk to the dog, or when there is a loss of deep pain perception for several days, conservative therapy may be indicated. The dogs must maintain a restricted regime in the cage for 4 weeks. Rest is essential, as there is a risk of further disc herniation if the dog is too active. This is also important when administering analgesics. Rehabilitation should be limited to passive range-of-motion exercises, neuromuscular electrical stimulation, and assisted standing outdoors during defecation for the first 2 weeks after surgical therapy [24,74]. The decision to use nonsurgical or surgical therapy is based on a number of factors, including financial considerations, as surgical therapy is usually more expensive. However, the nature of the disease, the degree of neurologic dysfunction, the severity of clinical symptoms, and the chronicity of the problem are the most important factors in choosing the form of therapy. If clinical symptoms persist or worsen after medical or conservative therapy, surgical intervention must be considered as soon as possible [75]. The benefit of early decompression in veterinary medicine has not been clearly demonstrated in prospective studies. However, from primary reports and meta-analyses in human medicine, there is evidence that early surgical decompression may shorten hospital stay and thereby reduce complications and also improve neurologic outcomes. A recent study by Martin et al. (2020) suggests that same-day surgery may reduce the risk of loss of pain perception in dogs with thoracolumbar disc extrusion [35]. Another study that looked at long-term neurologic outcome after hemilaminectomy and disc fenestration for therapy of canine IVDH between the years 2000 and 2007 found that 14.6% of dogs were unsuccessful and 85.4% of dogs were successful. There were 620 dogs that had intact nociception before surgery, and 97.7% of them were ambulatory after surgery [76]. Overall, however, the evidence for early surgical intervention remains weak in both human and veterinary medicine [77,78,79,80]. Through correlation analysis, the authors were unable to demonstrate the influence of body weight or the duration of clinical symptoms on the duration of recovery and overall outcome. The correlation coefficient only indicated a positive correlation between the neurologic dysfunction score and the time to first improvement after surgical therapy [81]. From the literature, the speed of onset of clinical signs is unlikely to affect the recovery of dogs with preserved deep pain perception and cannot be used to predict the prognosis of full return of locomotor function. Similarly, the results suggest that it is not possible to determine whether full recovery of function will occur and how long recovery will take based on the duration of clinical signs [81,82,83]. 

During the first study period, most diagnoses were based on radiographs and clinical examinations, and therapy was mostly conservative; however, surgical therapy was performed in 13% of dogs with progressive neurologic deficits. In 2015 to 2018, when MRI became more accessible than in the previous period, surgical therapy was performed in 24% of dogs compared with 66% cases undergoing conservative therapy. In the third period, between 2019 and 2022, surgical therapy was performed in 47.7% of dogs. With the increasing availability of MRI, the incidence of surgical therapy was higher. In their study, Nečas et al. (1999) reported a recurrence of clinical symptoms in surgically treated patients with thoracolumbar spine disease of 14.59% [50]. In our study, the recurrence rate averaged 59% with conservative therapy and 25% with surgical therapy. This higher incidence may be due to the fact that all segments were included, not only T3-L3 segment. 

Prognosis after spinal cord injury is usually mainly related to the loss or presence of deep pain perception. It is the best prognostic indicator for predicting prognosis in herniated intervertebral discs, but also after external spinal cord trauma [84]. During our evaluation period, we recorded 21 dogs without deep pain perception, of which 10 were euthanized immediately at the owner’s request and the other 6 were euthanized later because of persisting dysfunctions. The remaining five dogs lived long-term with manual bladder management in wheelchairs (unpublished data). Although young age is often associated with a better prognosis, there is limited evidence on the relationship between age and the time needed to learn to walk again. Several studies have examined whether changes in spinal cord parenchyma detected by MRI can predict a patient outcome after spinal cord injury. The results suggest that the contusion occurring at the time of trauma determines the severity of the lesion and prognosis, rather than the degree of compression [12,18,43,85].

## 5. Conclusions

The increasing availability of MRI over the years has increased the diagnostic and therapeutic success rate. Our results reflect changes in the population structure of the dogs studied and changes in the use of diagnostic methods that directly or indirectly influence the choice and success rate of therapy. These results may serve as a basis for further studies that look more closely at the details of injury in a larger group of dogs, and this information may be of interest to owners, breeders, and practicing veterinarians. 

## Figures and Tables

**Figure 1 vetsci-10-00359-f001:**
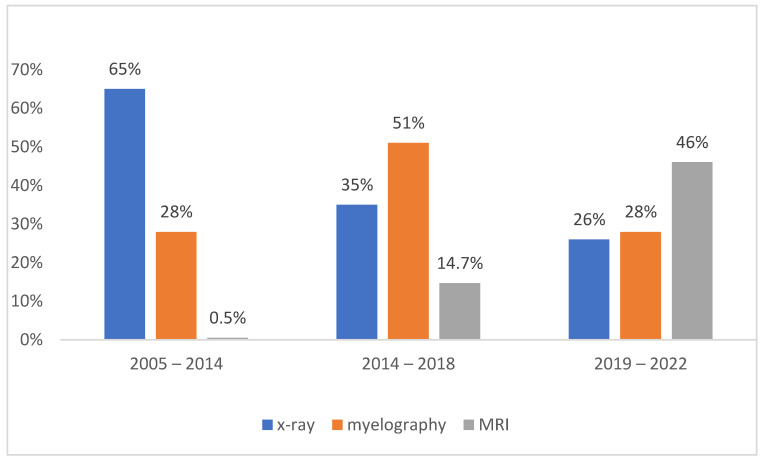
Frequency of diagnostic methods used in the three study periods.

**Figure 2 vetsci-10-00359-f002:**
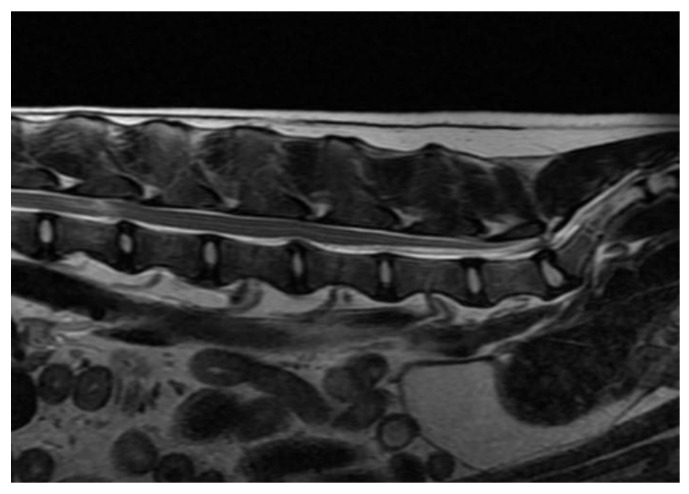
Anonymised T2w sagittal image from magnetic resonance imaging of a dog with acute, non-degenerated nucleus pulposus extrusion between L2 and L3 (with obvious intramedullary lesion just cranial to the intervertebral disc space), disc protrusion between L4 and L5, and evidence of degenerative lumbosacral stenosis with disc degeneration between L7 and S1.

**Figure 3 vetsci-10-00359-f003:**
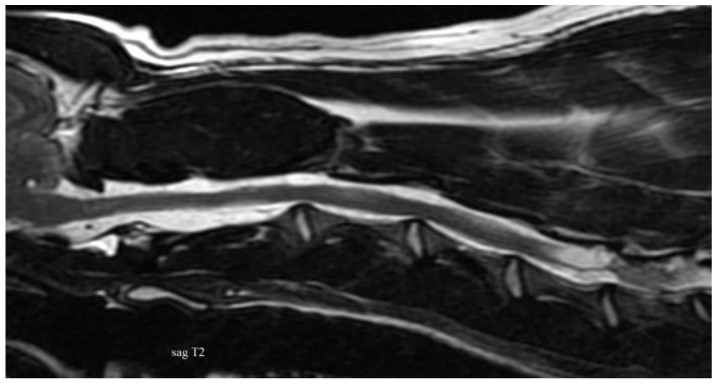
Anonymised T2w sagittal image from magnetic resonance imaging of a dog with cervical subarachnoid diverticulum at the level of C5 causing severe spinal cord compression.

**Figure 4 vetsci-10-00359-f004:**
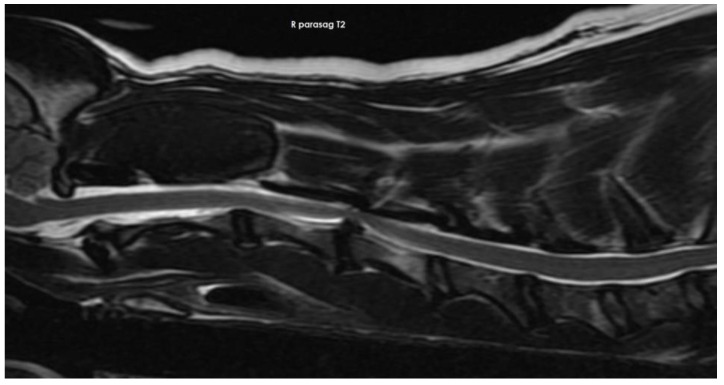
Anonymised T2w sagittal image from magnetic resonance imaging of a dog with intervertebral disc extrusion between C3 and C4.

**Figure 5 vetsci-10-00359-f005:**
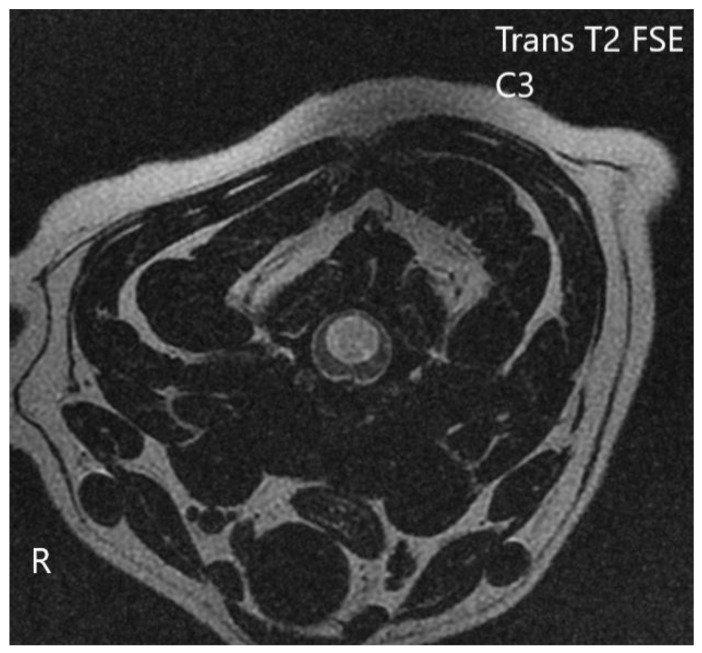
Anonymized T2 FSE transversal image from magnetic resonance imaging of a dog with syringohydromyelia in cervical segment causing severe spinal cord compression.

**Figure 6 vetsci-10-00359-f006:**
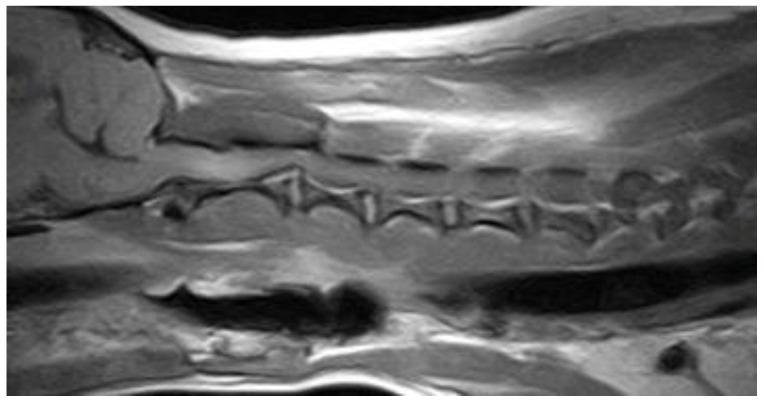
Anonymized T1w sagittal image from magnetic resonance imaging of a dog with steroid-responsive meningitis arteritis. Meningeal enhancement visible at the level of C2.

**Figure 7 vetsci-10-00359-f007:**
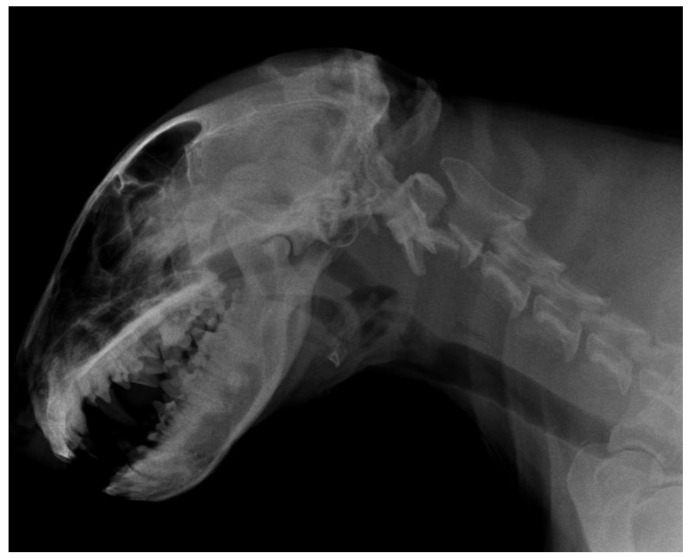
Anonymised radiographic finding of a dog with dens axis (C2) fracture.

**Figure 8 vetsci-10-00359-f008:**
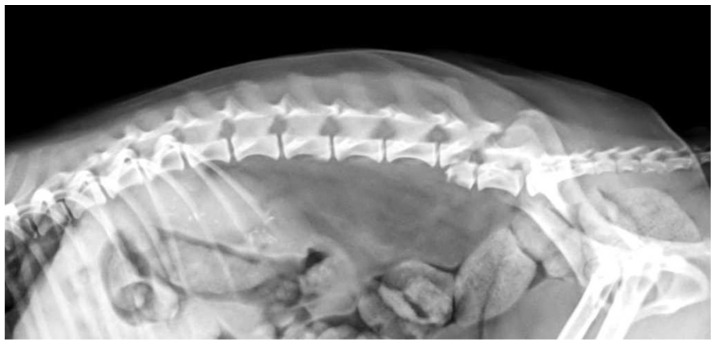
Anonymised radiographic finding of a dog with fracture of the sixth lumbar vertebra (L6).

**Figure 9 vetsci-10-00359-f009:**
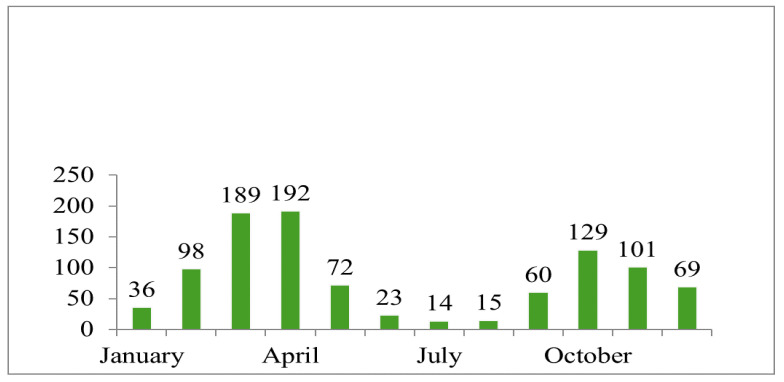
Number of dogs admitted with affected spine in each month.

**Table 1 vetsci-10-00359-t001:** Gender, age, and body weight description of the population of dogs evaluated in respective time periods. The same letter in superscript indicates which differences were statistically significant (*p* < 0.01).

	2005–2014 (*n* = 597)	2015–2018 (*n* = 204)	2019–2022 (*n* = 197)
Females/Males (Spayed/Castrated)	252/345 (191/242)	119/85 (81/63)	103/94 (72/52)
Age (months) Median (IQR) Min–Max	86 ^a^ 6–192	75 ^ac^ 7–169	88 ^c^ 10–158
Body weight (kg) Mean ± SD Min–Max	23.6 ± 14.02 ^ab^ 4.1–52.8	14.5 ± 12.02 ^a^ 4.7–38.9	17.5 ± 16.61 ^b^ 3.2–58.2

**Table 2 vetsci-10-00359-t002:** Representation of the most affected breeds over the evaluated time periods.

Most Affected Breeds	2005–2014 (*n* = 597)	Most Affected Breeds	2015–2018 (*n* = 204)	Most Affected Breeds	2019–2022 (*n* = 197)
German Shepherd Dogs	125 (20.9%)	Dachshunds	35 (17.1%)	French Bulldogs	34 (17.2%)
Dachshunds	85 (14.2%)	German Shepherd Dogs	30 (14.7%)	Yorkshire terriers	24 (12.1%)
Mixed-breed dogs	76 (12.7%)	Mixed-breed dogs	28 (13.7%)	Dachshunds	23 (11.6%)
Cocker Spaniels	42 (7%)	Cocker Spaniels	18 (8.8%)	Mixed-breed dogs	22 (11.1%)
Other breeds [number of different breeds]	296 (45%) [12]	Other breeds [number of different breeds]	93 (45.6%) [9]	Other breeds [number of different breeds]	94 (48%) [16]

**Table 3 vetsci-10-00359-t003:** Neuroanatomic localization of diagnosed lesions with indication of the most commonly affected breed for each segment.

Spinal Segment	2005–2014 (*n* = 597)	Most Affected Breed	2015–2018 (*n* = 204)	Most Affected Breed	2019–2022 (*n* = 197)	Most Affected Breed
C1-C5	46 (7.8%)	Jack Russel Terrier (16)	13 (6.3%)	Chihuahua (4)	49 (24.8%)	Yorkshire terrier (22)
C6-T2	14 (2.2%)	Pinscher (5)	8 (4%)	Lhasa Apso (3)	12 (6.2%)	Dachshund (3)
T3-L3	411 (68.8%)	Dachshund (56)	98 (48%)	Dachshund (28)	103 (52.3%)	French Bulldog (28)
L4-S3	126 (21.2%)	German Shepherd Dog (48)	83 (40.7%)	German Shepherd Dog (40)	33 (16.7%)	German Shepherd Dog (19)

**Table 4 vetsci-10-00359-t004:** Summary of diagnoses made in the dogs studied throughout whole study period, indicating the percentage of lesions or myelopathies visualized by two imaging methods. Some of the definitive diagnoses were confirmed by histopathology or laboratory tests, while some of the diagnoses were radiological diagnoses only, with no other confirmed cause of myelopathy.

Differential Diagnosis	Percentage of Affected Dogs	Diagnosis	Percentage of Diagnosis Gained with X-ray ± Myelography	Percentage of Diagnosis Gained with MRI
Vascular	5	Fibrocartilaginous thromboembolism		100
		Spinal cord haemorrhage		100
Inflammatory/infectious	12	Discospondylitis	80	20
		Steroid responsive meningitis arteritis		100
		Granulomatous meningoencephalomyelitis		100
		Toxoplasmosis		100
		Spinal empyema	25	75
		Osteomyelitis of vertebrae	80	20
Traumatic	15	Fractures	80	20
		Atlantoaxial subluxation	70	30
		Intervertebral disc extrusion	60	40
		Subdural/intraspinal haemorrhage		100
Anomalous	8	Subarachnoid diverticula		100
		Atlantoaxial subluxation	90	10
		Syringohydromyelia		100
		Hemivertebrae	100	
Metabolic/toxic	7	Thiamin deficiency		100
		Copper deficiency		100
		Organophosphate poisoning		100
Idiopathic	5	Diffuse idiopathic skeletal hyperostosis	100	
Neoplastic	8	Tumour of the vertebrae	80	20
		Lymphoma		100
		Hemangiosarcoma		100
Degenerative	40	Intervertebral disc degeneration	40	60
		Degenerative myelopathy		100
		Degenerative lumbosacral stenosis	80	20
		Caudal cervical spondylomyelopathy	40	60
		Spondylosis deformans	100	
		Osteoarthritis of facet joints	35	65

## Data Availability

Not applicable.

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
