# Peer review of "Comparison of Imaging Methods and Population Pattern in Dogs with Spinal Diseases in Three Periods between 2005 and 2022: A Retrospective Study"

_vetsci, 2023, doi:10.3390/vetsci10050359_

Round 1
Reviewer 1 Report
The authors did an excellent job of collecting data with a good number of patients considered. However, I do not agree with some statements regarding the therapeutic choice (lines 397-400): the medical point of view of the veterinarian is a different from the owner's point of view who certainly take into account the financial aspect. Furthermore, the choice of surgical therapy is closely linked to the type of pathology.
In my opinion, it is impossible to include small and large dogs in the same case study category as the type of disc pathology is strongly connected to the spatial relationship between the spinal cord and spinal canal, a different relationship in different breed.
Furthermore, traumas (fractures, dislocations of the spine, etc.) should not be considered because the treatments for these pathologies do not depend so much on the diagnostic methodology used as on a other pathological factors.
Reviewer 2 Report
"The paper submitted to the journal is quite interesting and provides additional information to the literature. However, there are some important information that is missed. Thus, it would be very valuable to know the specific age of the affected animal in each period. Moreover, since you are working with a quite large number of animals, it is mandatory to treat the data with a statistic treatment. Therefore, I recommend rejection and encourage resubmission with this information. Specific commentsIn the simple summary, please redone this paragraph since it is not so clear. In the introduction section, lines 27-28, what about neoplasias such as LSA, meningiomas or schwannomas. In the material and methods section The specialists were board certified neurologist? It would be very valuable to know the number of animals that underwent to MRI only in each period. In the statistical study, you do not explain the data treatment used. Line 197-197, which were the most affected breeds here? Line 205-206, same, similar than above, the affected breeds Are not clear. Line 272, this sentence should be included in another or distinct paragraph Line 277, this is the explanation of lower incidence, please clarify "
Reviewer 3 Report
Dear authors, the topic of the paper is interesting for the practitioners, and provides useful information.
However, the are several points to be addressed before considering the publication.
-In general, there is lack of description of the specific spine disorders diagnosed. There is mention to degenerative, inflammatory or neoplastic disorders without more specific information regarding the disease; i think that this aspect is of paramount importance to properly determine the real improvement of MRI as diagnostic tool, especially for intraparenchimal disorders not visible with other imaging methods.
-As far as it regards the introduction, a brief description of the most frequent spine disorders should be addressed, then describe the available literature regarding the topic and finally introduce the aims of the work. Information presented in this form result difficult to read and to understand.
-In the materials and method sections there is complete lack of techinal detailes regarding the imaging protocol; how mielography was performed? Regarding CT, only mielography was performed or there ara data regarding diagnosis performed with plain CT? High field or low field MRI was used, and which imaging protocol, coils ecc were used?
-As for the results, this section is rich in numbers and data, thus making the reading difficult; i think that the paper could benefit from dividing the data in tables, leaving in the text the most important results, thus making the results easier to understand.
-In this section, please provide more detailed information regarding specific neurodiagnosis. Degenerative, inflammatory or neoplastic is too vague and it is not clear if the use of MRI improved the diagnosis of specific disorders, it could be interesting to provide more detailed information regarding the percentage of specific diagnosis detected with each modality.
Then provide imaging to compare the different modalities, especially for the same patients examined with more than one imaging technique.
-For the discussion, the information presented in this form are hard to read and to understand; please, rephrase the section so to make the reading more fluid.
Reviewer 4 Report
Reviewed is the revised manuscript “Comparison of imaging methods and population pattern in dogs with spinal diseases in three periods between 2005 and 2022: a retrospective study” submitted by MVDr. Jakub Fuchs, et. al.
This study conducts a comprehensive long-term comparison of imaging modalities used for diagnosing neurological diseases in dogs related to the spine and spinal cord. The research spans three time periods (2005-2014, 2015-2018, and 2019-2022) and evaluates the occurrence of these diseases based on localization, gender, age, and breed. The paper reveals that the increased availability of magnetic resonance imaging (MRI) over the years has contributed to improved diagnostic and therapeutic success rates. It also highlights the changes in the population pattern of the evaluated dogs and the usage of different diagnostic methods, influencing therapy choice and success rates. The findings have significant implications for dog owners, breeders, practicing veterinarians, and insurance companies.
After considering the revisions suggested in the peer review, the paper is well-structured, informative, and relevant to its target audience. The authors have successfully addressed the concerns raised during the review process, and the manuscript is now ready for publication.
Author Response
Dear Reviewer,
we did our best to provide more scientific English language, and we removed the mentioned images.
Thank you!
The authors team

Round 2
Reviewer 1 Report
I take note the authors's replies
Author Response
Big thank you to the reviewer!
Reviewer 2 Report
This revised version of the paper shows an important improvement. However, the statistical questions are not addressed. Therefore, the study is fragile regarding the number of animals used (statistical significance, standard deviation and so on). I recommend checking this last point with a person with experience in statistical studies in large populations.
Author Response
Dear reviewer,
We tried our best to make this point addressed.
Thank you for your review:
Reviewer 3 Report
Dear authors,
thanks for replying to the comments.
I have few other comments to this version of the paper
line 14: CT abbreviation is missing
line 142: gadolinium dosage should expressed in mmol/kg, also details regarding gadolinium labeling are missing
paragraph 3.5 seasonal occurrence: this point is not clearly discussed in the following section. Please discuss this point, if it could be of particular interest compared to the available literature.
Finally i suggest to increase the number of the figures of the diseases discussed in the paper.
Author Response
Dear reviewer,
We did all revisions as you advised.
line 14: CT abbreviation is missing
The abbreviation CT is explained the first time when it is used in the text
line 142: gadolinium dosage should expressed in mmol/kg, also details regarding gadolinium labeling are missing
This information was added into the text.
paragraph 3.5 seasonal occurrence: this point is not clearly discussed in the following section. Please discuss this point, if it could be of particular interest compared to the available literature.
We added little more discussion concerning this point.
Finally I suggest to increase the number of the figures of the diseases discussed in the paper.
We added few more images of reported diagnoses.